# Effects of a Passive Back-Support Exoskeleton on Knee Joint Loading during Simulated Static Sorting and Dynamic Lifting Tasks

**DOI:** 10.3390/ijerph19169965

**Published:** 2022-08-12

**Authors:** Mona Bär, Tessy Luger, Robert Seibt, Julia Gabriel, Monika A. Rieger, Benjamin Steinhilber

**Affiliations:** Institute of Occupational and Social Medicine and Health Services Research, University Hospital of Tübingen, 72074 Tübingen, Germany

**Keywords:** knee force, tibiofemoral force, side effects, assistive device, asymmetric lifting, load shift, forward bent posture

## Abstract

Due to the load shifting mechanism of many back-support exoskeletons (BSEs), this study evaluated possible side effects of using a BSE on knee joint loading. Twenty-nine subjects (25.9 (±4.4) years, 179.0 (±6.5) cm; 73.6 (±9.4) kg) performed simulated static sorting and dynamic lifting tasks, including stoop and squat styles and different trunk rotation postures. Ground reaction force, body posture and the force between the chest and the BSE’s contact interface were recorded using a force plate, two-dimensional gravimetric position sensors, and a built-in force sensor of the BSE, respectively. Using these parameters and the subject’s anthropometry, median and 90th percentile horizontal (HOR_50_, HOR_90_) and vertical (VERT_50_, VERT_90_) tibiofemoral forces were calculated via a self-developed inverse quasi-static biomechanical model. BSE use had a variable effect on HOR_50_ dependent on the working task and body posture. Generally, VERT_50_ increased without significant interaction effects with posture or task. HOR_90_ and VERT_90_ were not affected by using the BSE. In conclusion, utilizing the investigated exoskeleton is likely to induce side effects in terms of changed knee joint loading. This may depend on the applied working task and the user’s body posture. The role of these changes in the context of a negative contribution to work-related cumulative knee exposures should be addressed by future research.

## 1. Introduction

Musculoskeletal disorders (MSD), especially in the back, remain the most common health problem affecting workers in the European Union [1] and the United States [2]. Twelve-month prevalence rates of 58% for the occurrence of general MSD [1] and of 25–43% for the back area have been reported [1,2]. Musculoskeletal back pain has been found to be strongly associated to physical risk factors, especially heavy lifting in the workplace and cumulative low back load. Therefore, intervention strategies for physically demanding work incorporating these risk factors need to remain a focus [2,3].

To support workers in their daily work routines, the use of exoskeletons has become a focus area. “Exoskeletons are assistive systems worn on the body that act mechanically on the body. In an occupational context, they aim to support functions of the skeletal and locomotor system during physical work” [4] (p. 3), by transferring forces from exposed body regions to other body sites [5]. Currently, one of the main debates about the use of exoskeletons is whether or not they are effective in preventing work-related MSD [4].

Recently, a growing number of studies have focused on biomechanical, physiological, and subjective stress and strain parameters for determining the impact of using exoskeletons on the musculoskeletal system during occupational tasks [6]. Passive back-support exoskeletons (BSEs) were shown to potentially reduce physical strain in the supported body area in experiments including dynamic tasks such as lifting [7,8,9,10,11] and in tasks with a static forward bent posture [12,13,14,15,16]. Describing the short-term influence of exoskeletons on physical stress and strain parameters in the supported body area is one important step toward identifying potential strategies for relieving these specific musculoskeletal structures in the wearer.

However, the nature of many exoskeletons is shifting the mechanical load from one area to a different area or areas of the body [5], which raises concerns about excessive biomechanical stresses on these other areas [4,17]. In this context, potential side effects of using BSEs have been examined using parameters such as muscle activity and perceived discomfort outside the target region [10,11,12,18,19,20,21,22,23]. With respect to side effects, findings for strain parameters in the legs, i.e., mean muscle activity and perceived discomfort have been inconsistent. Some studies report decreases [12,20,21,22], others report increases [10,11,18,19,20], or no statistically significant changes [10,11,12,18,19,23,24]. The ambiguous findings of the available studies and the fact that only few focused extensively on possible side effects in the leg region of using a BSE show that it is unclear whether and how using a BSE affects the musculoskeletal system of the lower limb [6].

To ensure the safe application of BSEs, including the aim of promoting workers’ health in physically demanding work, it is imperative to further investigate potential side effects (i.e., potential adverse consequences) of their use. An evaluation of biomechanical joint loading might also give more insight into load transfers or load shifts to other (i.e., non-supported or non-targeted) body areas caused by using a BSE [6,25]. Although the back, shoulder, and neck are much more commonly affected by MSD than the knee, Govaerts et al. (2021) reported a 33% overall prevalence of work-related MSD in the knee among industrial workers [26]. Moreover, there is reasonable evidence that knee disorders are related to physical work exposures partially similar to those reported for back pain, including awkward postures, lifting, and task repetition [1,27]. To the authors’ knowledge, so far there has been no published study focusing on the mechanical loading of lower limb joints, particularly the knees, when using a BSE. Hence, the aim of this study was to evaluate the horizontal (anteroposterior) and vertical forces acting on the tibiofemoral joints when using a BSE (Laevo^®^, Delft, The Netherlands) during simulated industrial work tasks. For this purpose, a self-developed two-dimensional inverse quasi-static biomechanical model was used. We hypothesized that the horizontal and vertical median and 90th percentile tibiofemoral forces increase when using the Laevo^®^ exoskeleton.

## 2. Materials and Methods

### 2.1. Sample Size and Study Design

This manuscript comprises one section of a broader, exploratory laboratory experiment, evaluating the effects of the Laevo^®^ V2.56 exoskeleton on physiological and biomechanical parameters using a within-subject-design [28] (registered at ClinicalTrials.gov, NCT03725982). A Single Williams Latin Square design [29] for six conditions ((1) *Exoskeleton*: Laevo^®^ exoskeleton (*EXO*) vs. *Control*; (2) *Task*: *Static* vs. *Dynamic*; (3) *Lifting style*: *Stoop* vs. *Squat*) was used to determine the sample size of 36 and to randomize the order of the main experimental tasks in this study. In addition, a Double Williams Latin Square design [29] was applied to randomize three *Trunk orientation* conditions for the tasks. The order of randomization resulted from drawing lots.

### 2.2. Participants

Thirty-nine male subjects were recruited to participate in the study, of which three subjects had to be excluded due to time restrictions (N = 1) or not meeting the BMI criterion (N = 2). Thirty-six healthy males completed the experiment, of which data from 29 subjects (mean age 25.9 (±4.4) years, mean body height 179.0 (±6.5) cm, mean body weight 73.6 ± 9.4 kg) were used for the outcome measures described here. The force plate data from seven subjects could not be used due to technical issues. Inclusion criteria were: male gender, age (18–40 years), BMI (18.5–30 kg/m^2^), and absence of any acute or cardiovascular diseases, physical disabilities, systemic diseases, or neurological impairments that would not allow subjects to perform the tasks or wear the exoskeleton. BMI was calculated by measuring body height and weight, while the other inclusion criteria were assessed according to subjects’ self-report. These restrictions in our study sample were chosen to avoid possible moderating influences of sex/gender, age, or even body composition, which have not previously been studied. Furthermore, male subjects were chosen due to the domination of males in the manufacturing industries. The Laevo^®^ exoskeleton is only adjustable to a restricted extent and might therefore not fit to all body dimensions (e.g., female body composition, BMI > 30 kg/m^2^) We chose a rather young age group to ensure that all subjects were able to perform the tasks without an early onset of fatigue.

The study was designed according to the Declaration of Helsinki and approved by the Ethics Committee of the University and University Hospital of Tübingen (617/2018BO2).

### 2.3. Exoskeleton

We evaluated the passive exoskeleton Laevo^®^ (V2.56, Laevo B.V., Delft, The Netherlands; 2.8 kg), which supports the back during work tasks such as lifting a load and tasks requiring forward bending postures. Torque generation is provided by two two-dimensional joints (“smart joints”) with gas pressure springs that are attached to a hip belt located close to the pivot point of the hip joints. Two rigid bars connect the joints to a chest pad placed over the upper part of the sternum and to two leg pads placed over the thighs. The smart joints can be turned on and off, and the joint flexion angle at which the support should begin can be set (range 0–45°, increments of 5°). The exoskeleton was adjusted to fit the subject’s physique in two ways: First, by varying the size of the exchangeable rigid bars connecting the chest pad and the smart joints resulting in a chest-to-smart joint distance of 405 mm (S-size) or 435 mm (L-size). Secondly, by adjusting the smart joint support angle to avoid contact forces while standing upright (depending on the subject’s torso composition). The force was measured and controlled using an integrated force sensor in the chest pad (38 × 10 mm; Type KM38-1kN, ME-Messsysteme GmbH, Henningsdorf, Germany). The leg-pad-to-smart joint distance could not be adjusted and was always 200 mm.

### 2.4. Experimental Procedure and Tasks

A 1.5-h visit to our laboratory was mandatory 1–5 days prior to participating in the experiment. This visit included information about the study procedure and signing an informed consent form. Inclusion and exclusion criteria were clarified, anthropometric measurements were collected, and subjects were familiarized with the exoskeleton and tasks. On the day of the experiment, which lasted 4 h, the subject was prepared with the measurement equipment required for the outcome measures and performed a series of experimental tasks [11,19,24]. This manuscript considers six experimental task conditions (*Static-EXO; Static-Control; Dynamic-EXO-Stoop; Dynamic-Control-Stoop; Dynamic-EXO-Squat; Dynamic-Control-Squat*; cf. Figure 1) and focusses on the outcome measures related to knee forces (i.e., tibiofemoral forces).

The experimental tasks were performed while standing on a force plate in front of a table that was adjustable according to the subject’s height. The feet position was defined prior to the experiment and kept constant during each task by using markings on the force plate (Figure 2). The feet position was defined while the subjects were instructed to stand comfortably upright with their feet positioned evenly and facing straight ahead. The distance to and height of the table was adjusted to allow the subject to perform the sorting or lifting task in the required body postures (explanation below). The six experimental conditions were performed in sets of three trials, with each trial performed in one of the three trunk orientations. Therefore, the sorting or the lifting box was placed to the front, in a 45° rotation to the left, or in a 45° rotation to the right from the sagittal plane. Reported results in the frontal direction include both knees when the work tasks were performed without trunk rotation. The reported ipsilateral results refer to both trunk orientations (left and right), including the knee belonging to the body side that coincides with the direction of trunk orientation. The reported contralateral results refer to both trunk orientations (left and right), including the knee belonging to the side of the body opposite to the direction of trunk orientation.

The simulated static work task included sorting screws and pins while keeping the trunk in a 40° forward bent posture in the sagittal plane, following the tangent line of a two-dimensional gravimetric position sensor (PS12-II; Thumedi GmbH & Co. KG, Thum, Germany) that was placed on the skin over the spinous process of the 10th thoracic vertebrae (T10). The examiner monitored the signal on a screen. Additionally, the subjects were instructed to almost completely extend but never overstretch their knees (stoop knee posture) and to keep their feet in the pre-marked position. The height of the table was adjusted and the y-position of the feet was set while the subjects remained in the forward bent posture, comfortably reaching the sorting material with their hands while their elbows were flexed at approximately 135°. The sorting task lasted 90 s without moving the feet, legs or trunk, and was performed in the two following conditions: with or without the exoskeleton (*Static-EXO* vs. *Static-Control*). The subjects rested for 30 s between each trunk orientation and for 120 s after each static experimental condition. (Figure 3a–c).

The simulated dynamic work task included lifting and lowering an 11.6 kg load (i.e., a 10 kg load placed into a 1.6-kg box [W × D × H of 60 × 40 × 22 cm] with handles on both sides [19 cm]). The pre-defined body posture for adjusting the table included bending the upper body at a 70° flexion-angle in the sagittal plane, controlled similarly to the static task, with the legs almost completely extended but not overstretched. The upper arms hung perpendicular to the platform with an elbow flexion of approximately 160° while holding the handles of the box. Each dynamic experimental condition consisted of two sets of five consecutive lifts, keeping a pace of 5 s per lift, timed by an acoustic signal. The subjects rested for 35 s between both sets. Each lifting repetition included the following movements: (1) starting in an upright standing position, bending the trunk forward and picking up the load; (2) resuming the upright position while holding the load close to the body in front of the pelvis with flexed elbows; (3) lowering the load by bending the trunk forward and returning the load to its original position; (4) resuming the initial upright standing position without the load. The lifting task was performed in the following four conditions: with or without the exoskeleton, and holding the knees almost extended (stoop style) or bending the knees (squat style) while lifting (*Dynamic-EXO-Stoop* vs. *Dynamic-Control-Stoop* vs. *Dynamic-EXO-Squat* vs. *Dynamic-Control-Squat*). The subjects rested for 60 s between each trunk orientation and for 180 s after each dynamic experimental condition (Figure 3d,e).

All tasks were approved for their work-related relevance by consulting seven industrial companies who were already testing or had interest in testing BSEs in their companies. The applied tasks and their executions, i.e., body postures, lifting frequency, working height, have best represented the real work situations of these consulted companies.

### 2.5. Measurement and Data Analysis

The outcome parameters to assess the forces acting on the tibiofemoral joints during the dynamic work task were 50th and 90th percentile horizontal (anteroposterior) forces (HOR_50_, HOR_90_), and 50th and 90th percentile vertical forces (VERT_50_, VERT_90_). They were considered as median and peak knee loads during the lifting tasks. During the static work task, only HOR_50_ and VERT_50_ were estimated, since the static body posture over the 90-s sorting task period would induce 90th percentile forces which do not differ much from the median forces. To estimate the forces acting on the knee joints, an inverse quasi-static model was developed, since no established model incorporating the Laevo^®^ exoskeleton that could be applied was available (see Appendix A for a detailed explanation of the model). Quasi-static models have been used previously to detect the risk of injury in industrial workers [30]. For the model, subjects’ anthropometrics, including body height, segment lengths, and segment weights [31,32,33], distances between the devices’ contact points, lower limb posture, ground reaction forces below the feet, and the force between the chest and the exoskeleton’s contact surface, were recorded.

To measure the lower limb posture, we used gravimetric inclination sensors connected to a sampling and storage device (PS12-II with 2.5D-gravimetrical sensors; THUMEDI GmbH & Co. KG, resolution 0.1° and 125 ms in time; maximum static error 0.5°; maximum repetition error 0.2°) attached to the skin over the anterior tibia and femur using double-sided adhesive tape (25 × 20 mm, 3M transparent Medical Standard, Top Secret^®^, Gesellschaft für Haarästhetik mbH, Fürth, Germany). The measurement system continuously recorded the anteroposterior and lateral inclination angles respective to the gravitational axis. Possible angular offsets caused by individual placement of the sensors at the tibia and femur were neutralized using the measurement values of a 5-s upright standing period recorded prior to the experiment.

Ground reaction forces were continuously recorded using a three-dimensional force plate that was linked to a signal conditioner and digitizer (FP9090-15-1000; Analog and Digital Amplifier AM6800; resulting resolution 0.5 N and 125 ms in time; overall maximum error 6 N; Bertec Corporation, Columbus, OH, USA). The resulting digital force signals were continuously recorded by self-developed software (University Hospital Tübingen) using the Bertec “Device interface Library for NET”, which allows recoded data to be synchronized with data captured by the inclination sensors placed on the lower limbs.

The force plate’s platform was prepared with a coordinate system to determine the subject’s standing position, which was kept constant for the static and dynamic tasks (see 2.4 Experimental procedure and tasks; Figure 2). The points forming the tangent between the lateral and medial malleoli were marked on the coordinate system and used for further calculations. Prior to each measurement session, a self-calibration procedure was executed to remove possible offsets, for example, caused by temperature variations. The position measurement accuracy was regularly checked by placing a 2 kg weight on five predefined locations on the force plate (at the center and close to the four corners); accuracy was accepted with measured location errors < 10 mm.

The support moment of the exoskeleton was estimated by measuring the contact force between the Laevo^®^ exoskeleton and the chest using a Ø38 mm × 10 mm thick force sensor (Type KM38-1kN, ME-Meßsysteme GmbH, Henningsdorf, Germany; resolution 0.1 N; maximum error 1% = 10 N, shown to be <2.5 N in this study setting) that was manually integrated in the chest pad of the exoskeleton and connected to the previously described sampling and storage device (PS12-II, 24 Bit physical resolution, 4096 Hz sampling rate).

Several of the subjects’ anthropometrics (i.e., body height, body weight), segment lengths, and distances (i.e., shank and thigh lengths, distances between sesamoid and malleolus) and segment distances of the exoskeleton (i.e., distance between joints and contact points) were included in the model for the force calculations (Cf. Appendix A).

### 2.6. Statistical Aanalysis

The normal distribution of the histograms of the outcome parameters was inspected visually and the absolute *z*-values of the skewness and kurtosis of the data were judged to be valid for statistical evaluation [34]. We used repeated-measures analyses of variance (RM-ANOVA) with fixed factors *Exoskeleton (E), Trunk orientation (TO)* and *(E × TO)* to analyze differences between the experimental conditions (1) *Static-EXO* vs. *Static-Control* for the outcome parameters HOR_50_ and VERT_50_. We used RM-ANOVA with fixed factors *E, TO, Lifting Style (LS), E × TO, E × LF, TO × LS, and E × TO × LS* to analyze differences between the experimental conditions (2) *Dynamic-EXO-Squat* vs. *Dynamic-Control-Squat*, and (3) *Dynamic-EXO-Stoop* vs. *Dynamic-Control-Stoop* for the outcome parameters HOR_50_, HOR_90_, VERT_50_, VERT_90_. However, only the findings including the *Exoskeleton*-condition are presented in the results section. To evaluate the static sorting task, we included the full 90-s periods. To evaluate the dynamic lifting task, we included only the two phases including the weight: (2) resuming the upright position while holding the load and (3) lowering the load by bending the trunk forward and returning the load to its original position. If statistically significant interaction effects occurred, Student’s *t*-tests were used for post-hoc pairwise comparisons. Further interpretations only considered the relevant comparisons (i.e., *EXO* vs. *Control* within each *Trunk orientation*: *ipsilateral*, *frontal*, *contralateral*, within each *Lifting style: Stoop, Squat*, and within the combination of *Trunk orientation* and *Lifting style*). For fixed effects, *F*-values, *p*-values, and effect size partial eta squared ($\eta_{p}^{2}$) were calculated using the *F*-ratios strategy [35], and for the post-hoc pairwise comparison, *T*-value, *p*-value, and effect size Cohen’s *d* were calculated using the pooled standard deviation strategy [36]. In agreement with Cohen [36] and *F*-ratios strategy [35], effect sizes were interpreted as small ($\eta_{p}^{2}$ ≤ 0.02; *d* ≤ 0.2), medium ($\eta_{p}^{2}$ 0.13–0.259; *d* 0.5–0.79), or large ($\eta_{p}^{2}$ ≥ 0.26; *d* ≥ 0.8). For pairwise comparisons, we accepted significance levels of α ≤ 0.05 for fixed effects, and of α ≤ 0.00333 for *E × TO*, α ≤ 0.00833 for *E × LS,* and α ≤ 0.00076 for *E × TO × LS* (Bonferroni correction for 15, 6, and 66 possible comparisons, respectively). JMP^®^ (Version 14.2.0, SAS Inc., Carry, NC, USA) was used for statistical evaluations.

## 3. Results

Median values with corresponding interquartile ranges (IQR) and differences between *EXO* and *Control* are provided in Table 1 for the main comparisons of the static and dynamic work tasks, in Table 2 for the *E × TO* and the *E × LS* comparisons for the static and dynamic work tasks, and in Table 3 for the *E × TO × LS* comparisons of the dynamic work task. The related statistics for the main effects of the *Exoskeleton* condition (*EXO* vs. *Control*) and the interaction effects for *E × TO, E × LS, and E × TO × LS* are provided in Appendix B (Table A5) for all examined work tasks. All relevant pairwise comparisons for variables with significant interaction effects are provided in Appendix B (Table A6) for static and dynamic work tasks.

### 3.1. Static Task

In the static work task, *Exoskeleton* had no significant main effect on HOR_50_. However, there was a significant interaction effect for *E × TO* (*p* < 0.001; $\eta_{p}^{2}$ = 0.496), including a significant pairwise comparison for the *contralateral* side (*p* < 0.001; *d* = −0.912), which showed a reduction when using the *EXO* (−126.4%) (Cf. Appendix B).

The main effect of *Exoskeleton* was significant for VERT_50_ (*p* = 0.011; $\eta_{p}^{2}\;$= 0.209); the acting force increased (1%) when using the *EXO*. There was no significant interaction effect for *E × TO* on VERT_50_ (Cf. Table 1 and Appendix B).

### 3.2. Dynamic Task

Performing the dynamic work task, *Exoskeleton* had a significant main effect on HOR_50_ (*p* = 0.012; $\eta_{p}^{2}=$ 0.205) with significant interaction effects for *E* × *TO* (*p* < 0.001; $\eta_{p}^{2}=0.455$), *E* × *LS* (*p* < 0.001; $\eta_{p}^{2}$ = 0.471), and *E* × *TO* × *LS* (*p* = 0.002; $\eta_{p}^{2}=$ 0.201). Pairwise comparisons for *E* × *TO* were significant only for *contralateral* (*p* < 0.001; *d* = −0.493). Pairwise comparisons for *E* × *LS* was significant only for *Squat* (*p* < 0.001; *d* = −0.261). Pairwise comparisons for *E* × *TO* × *LS* were significant for *E* × *ipsilateral* × *Squat* (*p* < 0.001; *d* = −0.195), for *E* × *frontal* × *Stoop* (*p* < 0.001; *d* = 0.597), for *E* × *contralateral* × *Squat* (*p* < 0.001; *d* = −0.487), and for *E* × *contralateral* × *Stoop* (*p* < 0.001; *d* = −0.717). (Cf. Appendix B) *EXO* decreased HOR_50_ when performing the task in *Squat style* in all directions (−61.1–−19.8%), and increased HOR_50_ when performing the *Stoop style* in *ipsilateral* and *frontal* (+13.2; +20.6%) and decreased HOR_50_ when performing the *Stoop style* in *contralateral* (−85.5%) (Cf. Table 3).

*Exoskeleton* had no significant main effect on HOR_90_. However, there was a significant interaction effect for *E* × *TO* (*p* < 0.001; $\eta_{p}^{2}$ = 0.292) and *E* × *LS* (*p* = 0.006; $\eta_{p}^{2}$ = 0.236), but without reaching statistical significance in the relevant pairwise comparisons and without interaction effects for the threefold interaction *E* × *TO* × *LS* (Cf. Appendix B).

*Exoskeleton* had a statistically significant main effect on VERT_50_ (*p* < 0.001; $\eta_{p}^{2}$ = 0.376) without any significant interaction effects (Cf. Appendix B). VERT_50_ slightly increased when using the *EXO* (≤ 7%) (Cf. Table 3).

Using the *EXO* had no significant effect on VERT_90_ (Cf. Appendix B).

### 3.3. Support Moment

Descriptive information about the 50th and 90th percentile support moment of the exoskeleton while performing the work tasks is provided in Table 4.

## 4. Discussion

Numerous studies have evaluated the use of occupational BSEs on short-term changes in physical stress and strain parameters in the body region supported by the exoskeleton. Only a few studies also investigated potential side effects of using occupational BSEs [6]. Therefore, the present study includes the evaluation of biomechanical knee joint loading when using a BSE. Using the Laevo^®^ exoskeleton had a variable influence on the anteroposterior acting horizontal forces, which seems to depend on the work task execution (e.g., lifting style) or posture (e.g., trunk orientation). Yet it remains unclear, whether the occurring changes are relevant in terms of knee joint health. Furthermore, vertical acting forces slightly increased due to the exoskeleton’s weight itself.

When performing the static sorting task in a forward bent static upper body posture with lateral trunk orientation, the ipsilateral knee was heavily loaded and the contralateral knee was almost unloaded. With respect to the horizontally acting forces on the femoral part of the knee joint, only the contralateral knee was significantly influenced by wearing the *EXO*. Without the *EXO*, the force mainly acted in anterior direction (*Static*-*Control-contralateral:* 20.7 ± 23.4 N), and with the *EXO* in a more posterior direction (*Static-EXO-contralateral*: −5.5 ± 48.3 N). The mechanical principle of transmitting load from the back to the leg pads via smart joints induced a translation force directed backwards onto the thighs [12], causing a posteriorly directed knee force.

Performing the dynamic work task using the *EXO* had an overall influence on HOR_50_. The major effect for work direction was observed on the *contralateral* side, reducing the anteriorly directed HOR_50_ for both *Lifting styles* (*Squat-contralateral*: −61.1%; *Stoop-contralateral*: −85.5%), while still being anteriorly directed (*Squat-EXO-contralateral:* 19.6 ± 69.5 N; *Stoop-EXO-contralateral*: 4.3 ± 58.6 N). Within the *E × LS* interaction, only the *Squat style* had a significant effect, reducing the anteriorly directed HOR_50_ even during frontally directed work and on the ipsilateral side during lateral work. In contrast, HOR_50_ tended to increase when performing *Stoop style* lifts during frontally directed work and during laterally directed work on the ipsilateral side, similar to our findings for the static work task. Both tasks were performed while maintaining almost extended knee postures compared to the *Squat style* (median flexion in *Static*: 19.7°; median and peak flexion in *Dynamic_Stoop*: 17.7°, 31.8°; median and peak flexion in *Dynamic_Squat:* 39.2°, 73.7° (0° flexion referring to fully extended knees)). Therefore, it is most likely that the effects of using the Laevo^®^ exoskeleton on horizontal knee forces depend on the wearer’s body posture (i.e., the knee flexion angle).

Shear force magnitudes and directions (anteriorly vs. posteriorly directed) have been shown to vary depending on knee flexion angles in isokinetic knee extension tasks [37,38]. As described in two associated publications [11,16], using the *EXO* led to more flexed knee joints in the static and dynamic tasks. The changes were most prominent in those tasks that included stoop postures (*Static*; *Dynamic_Stoop*), particularly *contralateral* (+95% in *Static*; +78.8% in *Dynamic_Stoop*) where we also detected most HOR_50_ changes when using the *EXO*. Accompanying the knee joint angle changes, the hip joints were also more flexed by the subjects when using the *EXO*, particularly in those tasks including stoop postures and observing the *contralateral* side [11,16]. Further, the support moment of the Laevo^®^ has been shown to depend strongly on the flexion angle of the smart joints which are located close by the hip pivot points [13]. Therefore, the leg pad pressure acting on the thighs must depend on the hip flexion angle, further influencing the horizontally acting knee forces.

Using the *EXO* had no effect on HOR_90_. It is likely that the EXO does not substantially alter peak horizontal forces when lifting and lowering a load in *Stoop* and *Squat Lifting styles*. Therefore, the Laevo^®^ presumably does not induce high peak horizontal loads on the knee joint. However, substantial time spent on knee straining work tasks, including those tasks without substantial force peaks (e.g., holding a posture), has been reported to be an important risk factor for musculoskeletal disorders in the knee [39,40]. Whether exoskeleton-induced changes in HOR_50_ can increase the risk for MSD is beyond the findings of the present study.

To our knowledge, there is no evidence on quantitatively reported knee forces in industrial work tasks and on potential changes induced by workplace interventions. Further, existing evaluations of knee forces, e.g., during daily activities, have been evaluated using different methods (i.e., in vivo measurements via telemetry, different biomechanical models) [41], which makes comparisons difficult. However, in previous studies, anteriorly directed knee forces evaluated during activities of daily living (i.e., walking, ascending and descending stairs, rising from or sitting down in a chair, single or two-legged stance) were reported to range from 0.04–1.6 times body weight (×BW) (peak) [37,42,43,44,45,46] and 0.09–0.18 × BW (mean) [43], and during squatting from 0.11–0.15 × BW (peak) and 0.02 × BW (mean) [42,43]. Posteriorly directed horizontal forces were reported to range from 0.23–1.7 × BW (peak) and 0.12–0.34 × BW (mean) [37,43,44] during daily activities, and from 0.2–3.6 × BW (peak) [37,47] during squatting. Neglecting bias due to insufficient comparability between methods and roughly approximating our data into a multiple body weight (×BW) metric (by dividing each measured force value [N] by the mean body weight of all included subjects (722.02 N)), we obtained the following forces when using the *EXO*: Anteriorly directed horizontal knee forces of ≤0.12 × BW (median) for static work tasks and ≤0.20 × BW (median), and ≤0.67 × BW (peak) for dynamic work tasks. Posteriorly directed horizontal knee forces of <0.01 × BW (median) only for the static task. This is within the force ranges reported for the common activities of daily living, although we included straining postures and additional loads. Therefore, it is possible that using the Laevo^®^ does not exert horizontal forces on the knee joints exceeding typical loads. However, the risk of developing degenerative MSD, such as osteoarthritis in the knee joint, has been shown to be related to cumulative loading over prolonged durations [40,48,49,50]. In Germany, osteoarthritis of the knee and meniscal lesions are listed as occupational diseases for which cumulative knee exposure is an important factor for their recognition [51]. In this context, future research should address a possible negative contribution of BSE use on cumulative loading of the knee joint.

Using the *EXO* had medium to large significant effects on VERT_50_. The force increased in the static work task by 0.9–13.7% (8.2–74.2 N) and up to 7.0% (58.1N) in the dynamic work task, without being influenced by *Trunk orientation* or *Lifting style. EXO* had no effect on VERT_90_. It can be assumed that the increases in VERT_50_ were mainly caused by the exoskeleton’s own weight (39 N; including the inbuilt force sensor), but probably not by any additional load transfer from the back to the legs.

Vertical acting knee forces have been reported to range between 1.0–10.0 × BW (peak) for activities of daily living [37,44,46,52,53,54,55], and to range between 0.3–5.6 × BW (peak) for squat tasks [37,47,54,55,56]. In the present study, when approximating the data into ×BW metrics, the vertical forces with using the *EXO* resulted in 0.48–1.25 × BW (median) for the static task, with 0.54–1.58 × BW (median), and 1.2–2.57 × BW (peak) for the dynamic work tasks. Similar to HOR, this lies within the range of the reported vertical forces when neglecting bias due to insufficient comparability between methods. However, in terms of cumulative knee loading as a risk factor for MSD of the knee, the weight of a BSE may represent a relevant additional load on the musculoskeletal system of the lower limbs. While the weight with 2.8 kg of the Laevo^®^ exoskeleton is rather light, other commercially available BSEs weigh up to 7 kg [57].

Until now, mainly muscle activity has been observed to estimate possible side effects of using BSEs [6]. The loading of the knee joints is highly influenced by forces exerted by the knee extensor and flexor muscles [58]. Therefore, it is likely that changes in occurring knee forces are accompanied by a changed activity of these muscle groups. Previous studies have reported that the knee extensor muscles are only slightly influenced by the use of the Laevo^®^ exoskeleton [9,11,16,18,59]. However, the activity of the gastrocnemius medialis muscle (GM) increased in all dynamic *contralateral* conditions, possibly due to postural changes [11], and the activity of the biceps femoris muscle (BF) decreased across all conditions as reported in two associated publications [11,16], possibly due to the supporting nature of the EXO for hip extension [11,12,16]. It has been reported that antagonistic coactivation of the hamstring is one important knee joint stabilizing factor which also influences forces acting on the knee joint [58]. Both GM and BF contribute to the antagonistic muscle activity with respect to the knee joint during the tasks observed here. Although a BSE may provoke changes in musculoskeletal strain (e.g., muscle activity) by addressing a specific joint (e.g., supporting hip extension), these changes also affect adjacent joints, such as the knee joint. In the present study, changing the activity of BF and GM by using the *EXO* may have produced secondary effects, such as changes of the knee joint forces. This is consistent with the assumptions of Park et al. (2022), who evaluated a BSE during walking and discussed an accompanying reduction in knee flexion torque along with a reduction in hip extension torque due to the hip extension support of the BSE, which may be caused by reduced BF muscle activity [60].

Although this was not explored further in the presented experiment, it is most likely that possible side effects are nearly proportional to the support provided by the device, which is caused by the load-transferring character of the BSE. According to our associated papers, the Laevo^®^ exoskeleton seems to provide a rather low back-relieving effect [11,16]. Consequently, side effects may also occur only slightly. Subsequently, using a BSE that provides a greater amount of support to the user may also cause more accompanying side effects. Further, mechanical differences of different devices may lead to different (side) effects due to the respective mechanical load transfer. For example, Alabdulkarim et al. (2019) compared three exoskeleton designs of upper body support exoskeletons during simulated overhead drilling. The findings demonstrated significant differences between the three exoskeleton designs in muscle activation of the supported area but also different muscle activation in non-target region which can be considered as side effects [61]. Therefore, before implementing a BSE, it is crucial to assess side effects that may occur for each individual device in its current version.

### 4.1. Limitations

Several limitations need to be address for the current study. First, the study population consisted of healthy male subjects aged 19 to 38 years, which does not reflect the general working population, also including female, aging, and physically impaired persons. Therefore, our results cannot be generalized. Second, seven out of originally 36 included subjects had to be excluded for the knee force calculations, resulting in a sample size of 29. Data had to be excluded due to technical issues while synchronizing the data for the first seven subjects. However, the body side that was prepared with the measurement equipment was still counterbalanced (in 14 subjects on the left, in 15 subjects on the right). Third, the Laevo^®^ exoskeleton is only partially adaptable to its wearer’s proportions. The distance between the hip joint and the chest was chosen between two available sizes (S, L), resulting in a lever arm of 405 mm or 435 mm. Only one of the 29 subjects used the S-sized model. The distance between the hip joint and the leg pad was not adjustable for the Laevo^®^, so the leg pads were not always placed exactly as specified by the manufacturer (i.e., lower than instructed for shorter subjects). Variations in exoskeleton placement on the body of the wearer can easily occur when such devices are applied in the field and must be minimized. Similarly, the exoskeleton cannot always be prevented from shifting during all movements. However, in our experiments, the fit of the Laevo^®^ was highly controlled by the examiners. Fourth, all subjects underwent a one-hour familiarization session, which might be too short to fully adapt to a routine exoskeleton use. Fifth, this experiment included three highly controlled simulated work tasks (e.g., on working posture). A real working environment, therefore, was not reflected, and possible variations of the exoskeletons’ effects are not known. Therefore, field studies under randomized, controlled conditions are needed to complement laboratory studies, which only provide initial insights into the acute possible effects of using an exoskeleton. Sixth, this study focused on acute effects of wearing an exoskeleton. Effects induced by regular long-term and full-shift use remain unclear and need to be investigated by long-term studies. Seventh, we used a self-developed biomechanical inverse quasi-static model for calculating moments and forces acting on the joints. The model includes some simplifications that might cause deviations from the actual occurring knee joint forces. Generally, in quasi-static models the dynamic movements are neglected [30] which could have biased the calculated forces in the dynamic lifting task. A comparison of a quasi-static vs. a dynamic model by Hariri et al., (2021) showed an underestimation of peak (19.7%) and cumulative spinal moments (3.6%) when not including the dynamic movements into the model in manual material handling tasks [30]. In particular, the 90th percentile knee forces could have been underestimated in this experiment. Further, only the vertical but not the horizontal components of the ground reaction force were included into the model. Some simplifications regarding the joint mechanics were adopted (i.e., neglecting torsional forces, assuming the pivot point being central and without shifting, treating joints like pure hinge joints, neglecting forces induced by antagonist muscles). The length of some body segments which were used for the model was estimated in relation to the respective body length. Only one leg was prepared with the measurement equipment (i.e., position sensors). Therefore, the force which was generated by the exoskeleton and acted onto the thighs was distributed onto both legs to be 50% loaded each. (Cf. Appendix A for detailed information about the model and its limitations.) Eighth, possible specific effects on the patellofemoral joint could not be assessed by our analysis. Those effects may be induced by the pressure of the exoskeleton’s pad onto the anterior upper leg muscles.

### 4.2. Key Points

The changes detected for HOR and VERT seem rather small and may not exceed typical ranges. However, it remains unclear what additional effect even small increases in acting knee joint forces have on musculoskeletal knee joint health, considering the contribution of cumulative loads to MSD of the knee.This evaluation shows that the side effects of using an exoskeleton depend on the work task executed (i.e., knee and trunk postures). Therefore, the decision to implement a BSE or not needs to depend on the individual work tasks.Back-support exoskeletons should be as light as possible, as their own weight seems to directly increase the vertical forces acting on the knee joint.Potential side effects, such as changes in knee joint forces, should be considered early in the development of a BSE.

## 5. Conclusions

When developing, evaluating, and applying a BSE, it is crucial to also focus on potential side effects that might occur when using the device during occupational tasks. We found task and posture-related changes in the loading characteristics of the knee joints when using the Laevo^®^ exoskeleton using our biomechanical model. Conclusions regarding the impact on musculoskeletal health risk for the knee would be beyond the present study. However, due to the cumulative nature of MSD, potential negative effects on the knee joints when using BSEs should be considered by future research.

## Figures and Tables

**Figure 1 ijerph-19-09965-f001:**
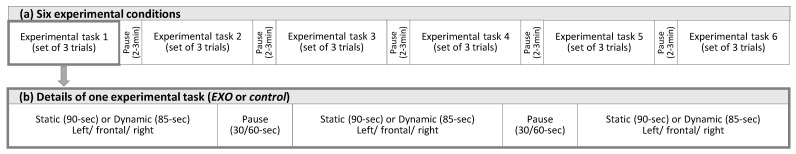
(**a**) shows the sequence of the six experimental conditions; two static and four dynamic: *Static-EXO; Static-Control; Dynamic-EXO-Stoop; Dynamic-Control-Stoop; Dynamic-EXO-Squat; Dynamic-Control-Squat.* The six conditions were performed in randomized order, and each was performed in a set of three *Trunk orientations.* Each set of static sorting tasks lasted 330 s, and each set of dynamic lifting tasks lasted 375 s. (**b**) shows one set of one experimental task. *Trunk orientations* (*left/frontal/right*) were performed in randomized order. Figure modified after Bär et al. (2022) [16].

**Figure 2 ijerph-19-09965-f002:**
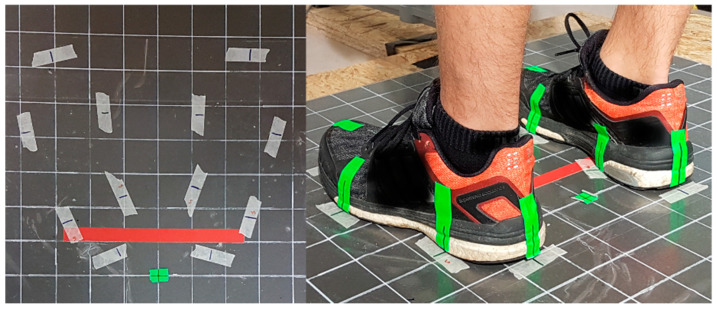
Force plate prepared with a coordinate system and with the individually pre-adjusted and pre-marked foot positions for the different tasks (static and dynamic). Marked landmarks were the heel in line with the Achilles tendon, medial and lateral malleolus, medial and lateral sesamoid, and the forefoot. Tape was placed on the subjects’ shoes and on the force plate, and a connecting line was drawn between each pair of foot-to-floor tape markings considering the above outlined landmark positions. The malleolus markers were later used to determine the x and y coordinates of the ankle joint centers for both feet; by calculating the midpoints of the lateral and the medial malleoli.

**Figure 3 ijerph-19-09965-f003:**
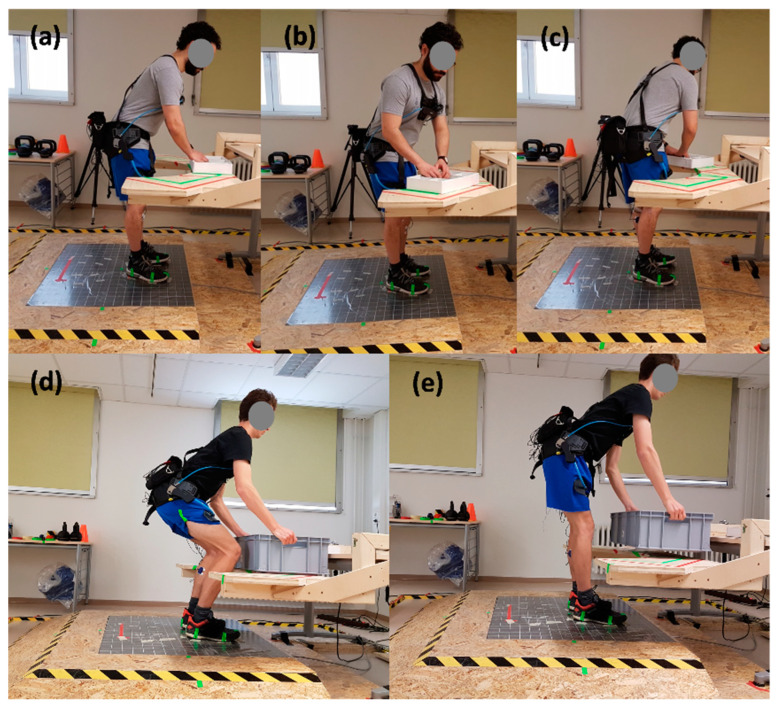
Subjects performing the experimental work tasks using the exoskeleton. (**a**–**c**) [16] show the static sorting task in three *Trunk orientation* conditions: (**a**) frontal, (**b**) right orientation, (**c**) left orientation. (**d**,**e**) show the dynamic lifting task to the front, performing (**d**) the squat style and (**e**) the stoop style.

**Table 1 ijerph-19-09965-t001:** Median knee force values and corresponding interquartile ranges (IQR), absolute and relative differences showing *EXO* compared to *Control* for static and dynamic work tasks (main interactions).

Work Task	Parameter	Knee Force *Control* [N]		Knee Force *EXO* [N]		Difference (*EXO-Control*)	
		Median	(IQR)	Median	(IQR)	[N]	%
Static	HOR_50_	49.69	(57.28)	46.45	(97.75)	−3.24	−6.5%
	VERT_50_	693.05	(527.15)	700.09	(478.77)	**7.04** ** ^μ^ **	**1.0%**
Dynamic	HOR_50_	52.71	(78.66)	36.56	(96.11)	**−16.15** ** ^μ^ **	**−30.6%**
	HOR_90_	251.91	(279.74)	246.99	(314.22)	−4.92	−2.0%
	VERT_50_	596.91	(376.75)	635.64	(375.45)	**38.74** ** ^λ^ **	**6.5%**
	VERT_90_	1010.14	(604.72)	1041.61	(599.21)	31.47	3.1%

Significant differences are shown in bold (*p*-value α ≤ 0.05). Effect sizes (^λ^ large effect size ($\eta_{p}^{2}$ ≥ 0.26); ^μ^ medium effect size ($\eta_{p}^{2}$ ≥ 0.13)) are shown for the significant differences. Detailed statistics are displayed in Appendix B. N = newton; HOR_50_ = 50th percentile of the horizontal force; HOR_90_ = 90th percentile of the horizontal force; VERT_50_ = 50th percentile of the vertical force; VERT_90_ = 90th percentile of the vertical force.

**Table 2 ijerph-19-09965-t002:** Median knee force values and corresponding interquartile ranges (IQR), absolute and relative differences showing *EXO* compared to *Control* for static and dynamic work tasks (two-fold interactions (**a**) *EXO × Trunk orientation* and (**b**) *EXO × Lifting styl**e*).

(a)			Knee Force *Control* [N]		Knee Force *EXO* [N]		Difference (*EXO-Control*)	
Work Task	Parameter	Trunk Orient	Median	(IQR)	Median	(IQR)	[N]	%
Static	HOR_50_	ipsi	57.63	(67.71)	60.11	(122.75)	2.48	4.3%
		front	73.32	(44.90)	88.02	(70.05)	14.69	20.0%
		cont	20.73	(23.37)	−5.47	(48.34)	**−26.19** ** ^λ^ **	**−126.4%**
	VERT_50_	ipsi	896.48	(423.04)	904.65	(433.03)	8.17	0.9%
		front	765.16	(202.38)	839.40	(232.69)	74.24	9.7%
		cont	307.59	(168.52)	349.64	(173.52)	42.05	13.7%
Dynamic	HOR_50_	ipsi	67.12	(96.00)	56.41	(109.21)	−10.70	−15.9%
		front	69.63	(84.01)	59.84	(105.20)	−9.78	−14.1%
		cont	29.84	(41.02)	3.53	(53.79)	**−26.30** ** ^σ^ **	**−88.2%**
	HOR_90_	ipsi	365.43	(326.53)	371.04	(346.22)	5.62	1.5%
		front	287.50	(178.97)	309.12	(224.50)	21.62	7.5%
		cont	95.25	(115.99)	78.93	(110.65)	−16.32	−17.1%
	VERT_50_	ipsi	767.35	(418.68)	806.37	(413.84)	39.03	5.1%
		front	652.35	(313.27)	696.94	(300.28)	44.59	6.8%
		cont	409.13	(243.75)	421.87	(233.23)	12.74	3.1%
	VERT_90_	ipsi	1406.16	(745.10)	1439.82	(723.97)	33.66	2.4%
		front	1009.79	(529.94)	1057.44	(499.05)	47.65	4.7%
		cont	798.45	(306.43)	809.17	(306.52)	10.72	1.3%
**(b)**			**Knee force *Control*** **[N]**		**Knee force *EXO*** **[N]**		**Difference** **(*EXO-Control*)**	
**Work Task**	**Parameter**	**Lifting Style**	**Median**	**(IQR)**	**Median**	**(IQR)**	**[N]**	**%**
Dynamic	HOR_50_	Squat	90.30	(107.91)	61.75	(112.25)	**−28.55** ** ^σ^ **	**−31.6%**
		Stoop	71.15	(102.42)	75.53	(149.61)	4.39	6.2%
	HOR_90_	Squat	301.63	(338.34)	261.76	(358.53)	−39.87	−13.2%
		Stoop	274.26	(286.83)	303.18	(349.62)	28.91	10.5%
	VERT_50_	Squat	613.48	(354.58)	653.30	(376.70)	39.82	6.5%
		Stoop	822.45	(681.98)	840.92	(658.81)	18.47	2.2%
	VERT_90_	Squat	1006.89	(451.76)	1054.35	(481.07)	47.45	4.7%
		Stoop	1343.11	(823.42)	1322.57	(771.82)	−20.53	−1.5%

Significant differences for the post hoc analyses are shown in bold (*p*-values α ≤ 0.00333 for *E* × *TO* and α ≤ 0.00833 for *E* × *LS*). Effect sizes (**^λ^** large effect size (*d* ≥ 0.8); **^σ^** small effect size (*d* ≥ 0.2)) are shown for the significant differences. Detailed statistics are displayed in Appendix B. N = newton; Trunk Orient = *Trunk orientation;* HOR_50_ = 50th percentile of the horizontal force; HOR_90_ = 90th percentile of the horizontal force; VERT_50_ = 50th percentile of the vertical force; VERT_90_ = 90th percentile of the vertical force; ipsi = *ipsilateral*; front = *frontal*; cont = *contralateral*.

**Table 3 ijerph-19-09965-t003:** Median knee force values and corresponding interquartile ranges (IQR), absolute and relative differences showing *EXO* compared to *Control* for the dynamic work task (three-fold interactions *EXO × Trunk orientation × Lifting style*).

Parameter	Lifting Style	Trunk Orient	Knee Force *Control* [N]		Knee Force *EXO* [N]		Difference (*EXO-Control*)	
			Median	(IQR)	Median	(IQR)	[N]	%
HOR_50_	Squat	ipsi	126.78	(142.73)	101.62	(146.48)	**−25.15**	**−19.8%**
		front	101.04	(91.74)	74.35	(92.33)	−26.69	−26.4%
		cont	50.36	(65.39)	19.57	(69.46)	**−30.79** ** ^σ^ **	**−61.1%**
	Stoop	ipsi	94.59	(117.18)	107.05	(159.15)	12.46	13.2%
		front	118.36	(90.48)	142.74	(124.94)	**24.39** ** ^μ^ **	**20.6%**
		cont	29.43	(33.49)	4.26	(58.56)	**−25.17** ** ^μ^ **	**−85.5%**
HOR_90_	Squat	ipsi	509.26	(462.72)	481.07	(404.54)	−28.19	−5.5%
		front	284.71	(173.79)	258.84	(238.40)	−25.87	−9.1%
		cont	151.54	(232.13)	109.48	(160.99)	−42.06	−27.8%
	Stoop	ipsi	361.67	(233.48)	406.50	(313.69)	44.83	12.4%
		front	346.07	(133.70)	392.77	(169.43)	46.70	13.5%
		cont	70.81	(67.34)	60.19	(88.21)	−10.61	−15.0%
VERT_50_	Squat	ipsi	833.72	(360.54)	891.83	(325.38)	58.10	7.0%
		front	623.58	(225.56)	667.24	(236.27)	43.66	7.0%
		cont	393.07	(290.35)	387.06	(261.02)	−6.00	−1.5%
	Stoop	ipsi	1125.73	(771.99)	1142.44	(708.87)	16.72	1.5%
		front	1006.42	(431.70)	1022.66	(423.27)	16.24	1.6%
		cont	413.25	(324.03)	439.88	(323.31)	26.63	6.4%
VERT_90_	Squat	ipsi	1319.08	(557.25)	1387.66	(541.02)	68.58	5.2%
		front	933.70	(363.33)	976.06	(373.88)	42.36	4.5%
		cont	864.42	(365.11)	868.97	(335.23)	4.56	0.5%
	Stoop	ipsi	1912.96	(760.15)	1858.51	(796.44)	−54.45	−2.8%
		front	1396.92	(510.29)	1399.91	(467.59)	2.99	0.2%
		cont	856.06	(364.26)	868.64	(359.74)	12.58	1.5%

Significant differences of the post hoc analyses are shown in bold (*p*-value α ≤ 0.00076 for *E × TO × LS*). Effect sizes (**^μ^** medium effect size (*d* ≥ 0.5); **^σ^** small effect size (*d* ≥ 0.2)) are shown for the significant differences. Detailed statistics are displayed in Appendix B. N = newton; *Trunk Orient* = *Trunk orientation;* HOR_50_ = 50th percentile of the horizontal force; HOR_90_ = 90th percentile of the horizontal force; VERT_50_ = 50th percentile of the vertical force; VERT_90_ = 90th percentile of the vertical force; ipsi = *ipsilateral*; front = *frontal*; cont = *contralateral*.

**Table 4 ijerph-19-09965-t004:** Median values and corresponding interquartile ranges (IQR) showing the support moment provided by the exoskeleton.

Support Moment [Nm]	Trunk Orient	Static Task		Squat Lifting		Stoop Lifting	
		Median	(IQR)	Median	(IQR)	Median	(IQR)
50th Percentile	ipsi	22.72	(7.26)	19.94	(13.21)	19.05	(16.21)
	front	23.24	(4.81)	20.79	(15.19)	20.93	(16.90)
	cont	22.72	(7.26)	19.94	(13.21)	19.05	(16.21)
90th Percentile	ipsi	NA	NA	29.15	(11.97)	30.25	(10.43)
	front	NA	NA	32.04	(10.61)	32.23	(11.09)
	cont	NA	NA	29.15	(11.97)	30.25	(10.43)

Trunk Orient = *Trunk orientation*; Nm = Newtonmeter; ipsi = *ipsilateral*; front = *frontal*; cont = *contralateral*.

## Data Availability

The data are not publicly available due to data use restrictions contained in study participants’ information material.

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
