# Peer review of "Effects of a Passive Back-Support Exoskeleton on Knee Joint Loading during Simulated Static Sorting and Dynamic Lifting Tasks"

_ijerph, 2022, doi:10.3390/ijerph19169965_

Round 1

Reviewer 1 Report

Effects of Wearing a Passive Back Exoskeleton on Knee Joint Loading during a Simulated Static Sorting and Dynamic Lifting Task (Manuscript ID: IJERPH-1787285)

General Comments:

The findings of this study are novel and informative as they tried to explore the possible side effects of a passive lift-assistive exoskeleton during static sorting and dynamic lifting tasks in terms of changing the knee joint loading. The authors used a self-developed inverse quasi-static biomechanical model to compute the median and 90th percentile horizontal and vertical tibiofemoral forces. The results indicated that BSE use showed variable effect on median horizontal tibiofemoral forces (dependent upon task and body posture). Also the findings revealed that median vertical tibiofemoral forces increased on average by wearing the exoskeleton. Interestingly, 90th percentile horizontal and vertical tibiofemoral forces did not alter while EXO was used. In sum, the authors mentioned that the exoskeleton induced some side effects in terms of altering the knee joint loading. Overall, the manuscript was almost clear and well-explained. However, several places in the manuscript need additional clarifications and/or modifications (see the comments for the details). Below, please find the feedback/suggestions to improve the quality of the manuscript.

Title:

I would suggest to change “Back Exoskeleton” to “Back-Support Exoskeleton” to be more consistent with the literature. I think your title reads better if you remove “wearing”?!

“a Simulated Static Sorting and Dynamic Lifting Task” --> “Simulated Static Sorting and Dynamic Lifting Tasks”

Abstract:

“Back-supporting” --> “Back-support”

The BSE in the first line should be “BSEs” as you have exoskeletons!

I would suggest to add the mean (SD) of age, weight, and height in the abstract.

You have more than one lifting task, therefore, remove a before dynamic lifting task.

Different lower limb might be a little bit unclear! Are you trying to say stoop and squat? If yes, you can modify the sentence and mention it clearly.

Line 18: “invers” --> “inverse”

Lines 18 & 19”: “Utilizing the BSE” --> “BSE use”

It’s okay to have several keywords but it seems that your current number of keywords is larger than the suggested number of keywords by IJERPH. You can remove one or a few of them!

1.     Introduction:

The introduction is easy to follow but I provided a few recommendations to further improve it.

Line 38: By saying “technical aids”, do you mean assistive devices? I think technical aids is very general and vague expression.

Lines 46 & 47: “the impact on musculoskeletal system of using exoskeletons in work tasks” --> “the impact of exoskeletons on the musculoskeletal system during occupational tasks”

Line 47: “back supporting” --> “back-support”

It would be great if you could add a few sentences about the advantages of passive exoskeletons vs. active ones. Additionally, the readers might be interested in knowing why currently more biomechanical evaluations are performed on passive EXOs and why past interventions were not fully effective (see the introduction in the following papers or you can cite them).

Alemi, et al. (2020). Effects of two passive back-support exoskeletons on muscle activity, energy expenditure, and subjective assessments during repetitive lifting. Human factors62(3), 458-474.

Alemi, et al. (2022). Modeling the metabolic reductions of a passive back-support exoskeleton. Journal of Applied Physiology132(3), 737-760.

Line 49: You can also cite the following recently published paper for dynamic tasks like lifting:

Alemi, et al. (2022). Modeling the metabolic reductions of a passive back-support exoskeleton. Journal of Applied Physiology132(3), 737-760.

Lines 59 & 60: I think the expression “studies report positive side effects” is misleading as decreasing strain in the legs is great and is not counted as positive side effect. You can remove this expression or make the whole sentence more clear!

Line 62: “results” --> “findings”

Line 67: the expression following i.e. can be placed in the parenthesis.

Line 74: “lifting” --> “lifting style”    &   “repetition” --> “task repetition”

Although your study is an exploratory study, I strongly suggest you to add your hypothesis in the last paragraph of the introduction (maybe in the last couple of sentences of intro).

2.     Materials and Methods:

It is really great that you included section 2.1 in the manuscript. Many papers are missing sample size justification section.

2.2. Participants

What is the main reason that you only had male subjects?

Lines 99 & 100: suggest to change the expression in the parenthesis to mean (±SD) of age, ….

Line 102: Make the inclusion criteria about age and BMI more clear!

2.5. Experimental Procedure and Tasks

I think instead of calling experimental task 1 to 6 in Figure 1, you can name the conditions (e.g., Static-EXO-Stoop). Parts a and b of Figure 1 look redundant. You can explain the figure’s content only inside the manuscript text.

Notice that you only have three trunk orientations for sorting task. For lifting, you only have two lifting styles. Therefore, Figure 1 is confusing. You need to also modify the sentence in Lines 162 & 163 as you don’t have three trunk orientations during lifting tasks.

Is there any specific reason that you selected forward flexion angle as 40º during sorting tasks? If yes, please justify it in the manuscript. Is there any reason that you selected 90 seconds for static sorting tasks? Why not 60 seconds? Did you select the sorting time based on your pilots?

Although I realize that it takes a few hours to finish the data collection for experiments similar to your study, I believe participants should be awarded more rest between the static sorting tasks as the task is somehow demanding. If you’ve had more than one trunk angles, the subjects would be fatigued with only 30 s rest between each trial.

Why you selected 5 sec per lift for dynamic lifting tasks? That’s a very short time increment! In addition, why the subjects had to bend the trunk forward for approximately 70 degrees. When you have a very controlled experiments, it is very hard to generalize your results to activities of daily living or other real-word tasks. Please explain this carefully.

Lines 206 and 207: You either remove the expression in the parenthesis (as it is redundant) or put comma instead of vs. in the parenthesis).  

2.5. Measurement and Data Analysis

Would you please explain why you did not select mean knee loads instead of median? Did you have skewed knee loading data? I am asking you because according to section 2.6, it seems that you checked the normality of the data visually!!! If the data is normal, why you don’t report the mean knee load? Furthermore, 90th percentile vertical force doesn’t show the peak knee load (however it might be close to peak)?! If you are planning to assess the risk of injury, it is better that you either select 95th percentile or the actual peak value (unless the peak value is much higher than 95th percentile).

2.6. Statistical Analysis

Line 265: You investigated the main effect of Exoskeleton and interaction effect of Exoskeleton × Trunk Orientation. In the text and for clarity, please explain why you didn’t include the fixed effect of Trunk Orientation in your model.

Why you selected student t-test for post-hoc analysis? Why not Tukey-HSD? Tukey-HSD seems more robust than student t-test!

Line 284: Please correct the confusing definition (i.e., definition in the parenthesis) of effect sizes for small, medium, and large! For example, for medium, d should be between 0.2 and 0.5!

3.     Results:

3.1. Static Task

Line 333: Please remove the two plus signs that are in the parenthesis.

3.3. Support Moment

Instead of having dash in Table 5, I would suggest to put NA. Additionally, please correct the caption of Table 5. 50th percentile shows the median and not the mean value. In the caption, you can also explain why 90th percentile for static tasks is NA.

4.     Discussion:

Thanks for your detailed and precise discussion. However, one of the main concern that needs to be addressed in this section is whether if using Laevo has changed the kinematics of lifting?! Have you ever checked some of the joint angles to see if the exoskeleton does not significantly change the lifting kinematic? Perhaps, Laevo doesn’t significantly change the kinematic during sorting tasks (due to the intrinsic nature of the task) but some of the exoskeletons might encourage the wearer to perform specific type of lifting (e.g., stoop or squat lifting). Have you done any kinematic analysis for your performed tasks? If yes, you can bring it up in the discussion section.

Line 405: “Stooped” --> “Stoop”

Lines 410 to 412: Make your sentence shorter and more clear!

Line 415: it?

4.1. Limitations

The limitations have been discussed clearly. However, the modeling simplifications/limitations could be elaborated more.

Line 501: “Some limitations need to be mentioned.” --> “Several limitations need to be address for the current study.”

Line 502: “19-38” --> “19 to 38”

Line 503: Please modify this sentence for clarity!

Line 513: Several exoskeletons have this problem, particularly during asymmetric tasks!

Lines 518-520: Although the adaptation time to the exoskeleton is not clear at this stage (i.e., dependent upon many factors), I am glad that you had separate training session to minimize the confounding effects due to the lack of proper adaptation to the EXO.

Lines 520-524: What was the main reason that you had high control for this experiment? Did you only want to keep the kinematic similar for the fair comparison? I think you could instruct the subjects to perform a specific task but let them to perform those tasks with slight variations. Otherwise, it will be very hard to generalize your findings to real-world scenarios.

Line 527: “invers” --> “inverse”

Reviewer 2 Report

The paper is generally well organized. Grammar usage is good. The paper is particularly well written. There are couple of sentences which reduce readability and ease of understanding, thus, it would be good that manuscript is proofread very carefully.
The paper reports what authors have done. There is not enough information about advantages of the methods used in the paper over the other methods in the literature. The authors must clearly position their paper within the context of relevant papers published in the field previously, ie, in terms of evaluation of  the horizontal (anteroposterior) and vertical forces acting during simulated industrial work.

Round 2

Reviewer 1 Report

I would like to thank the authors for addressing my comments and concerns. With their additional analyses and additions to the text, I am happy with the work in its current form.